# Downregulation of Astrocytic Kir4.1 Potassium Channels Is Associated with Hippocampal Neuronal Hyperexcitability in Type 2 Diabetic Mice

**DOI:** 10.3390/brainsci10020072

**Published:** 2020-01-30

**Authors:** Miguel P. Méndez-González, David E. Rivera-Aponte, Jan Benedikt, Geronimo Maldonado-Martínez, Flavia Tejeda-Bayron, Serguei N. Skatchkov, Misty J. Eaton

**Affiliations:** 1Department of Biochemistry, Universidad Central del Caribe, Bayamón, PR 00960-6032, USA; miguel.mendez3@upr.edu (M.P.M.-G.); ftejedabayron@gmail.com (F.T.-B.); 2Department of Sciences and Technology, Antilles Adventist University, Mayaguez, PR 00680, USA; 3Department of Natural Sciences, University of Puerto Rico, Aguadilla, PR 00604-6150, USA; 4Departments of Physiology and Biochemistry Universidad Central del Caribe, Bayamón, PR 00960-6032, USA; janbenedikt79@yahoo.com; 5School of Chiropractic, Universidad Central del Caribe, Bayamón, PR 00960-6032, USA; geronimo.maldonado@uccaribe.edu

**Keywords:** diabetes, hyperexcitability, astrocytes, Kir4.1, hippocampus, potassium uptake

## Abstract

Epilepsy, characterized by recurrent seizures, affects 1% of the general population. Interestingly, 25% of diabetics develop seizures with a yet unknown mechanism. Hyperglycemia downregulates inwardly rectifying potassium channel 4.1 (Kir4.1) in cultured astrocytes. Therefore, the present study aims to determine if downregulation of functional astrocytic Kir4.1 channels occurs in brains of type 2 diabetic mice and could influence hippocampal neuronal hyperexcitability. Using whole-cell patch clamp recording in hippocampal brain slices from male mice, we determined the electrophysiological properties of stratum radiatum astrocytes and CA1 pyramidal neurons. In diabetic mice, astrocytic Kir4.1 channels were functionally downregulated as evidenced by multiple characteristics including depolarized membrane potential, reduced barium-sensitive Kir currents and impaired potassium uptake capabilities of hippocampal astrocytes. Furthermore, CA1 pyramidal neurons from diabetic mice displayed increased spontaneous activity: action potential frequency was ≈9 times higher in diabetic compared with non-diabetic mice and small EPSC event frequency was significantly higher in CA1 pyramidal cells of diabetics compared to non-diabetics. These differences were apparent in control conditions and largely pronounced in response to the pro-convulsant 4-aminopyridine. Our data suggest that astrocytic dysfunction due to downregulation of Kir4.1 channels may increase seizure susceptibility by impairing astrocytic ability to maintain proper extracellular homeostasis.

## 1. Introduction

Epilepsy, characterized by recurrent seizure episodes, is one of the most common neurological disorders [1]. However, seizures may also be provoked by electrolyte imbalance such as hyponatremia, by metabolic disturbances such as hypoglycemia or hyperglycemia, as well as by high fever and head trauma [2]. Interestingly, 25% of patients with type 2 diabetes develop seizures throughout their lives [3]. It has been reported that non-ketotic hyperglycemia patients suffered from seizures with an unknown cause. These seizure episodes could often be reversed by insulin or sulphonylurea compounds, indicating that hyperglycemia is a great candidate causing this phenomenon [4,5]. In addition, a recent 10 year population-based study concluded that the incidence of epilepsy was 1.50 times higher in a cohort with type 2 diabetes as compared to matched controls [6]. Although provoked seizures can be caused by both hypoglycemia and hyperglycemia [4,5,7], our main focus in this study is on hyperglycemia and the potential role of astrocytes in seizure susceptibility. In respect to the focus of the present study, it is noteworthy that hyperglycemic diabetic animal models display synaptic transmission issues such as increased latencies of visual and auditory evoked potentials, hearing problems, impaired long-term potentiation and depression in hippocampal neurons [8,9,10]; all with yet unknown mechanism(s).

Astrocytes are one of the major cell types in the CNS and they enwrap blood vessels and neuronal synapses. These cells are imperative for proper brain function; particularly for K^+^ and glutamate homeostasis [11,12,13,14]. Inability of astrocytes to control K^+^ and glutamate levels, in active synaptic areas, may contribute to epilepsy and seizures [13,14,15]. Astrocytes have a hyperpolarized membrane potential mainly due to the expression of inwardly rectifying potassium channels 4.1 (Kir4.1) that provide the major K^+^ conductance [11,12,13,16,17]. These channels play a fundamental role in buffering excess K^+^ released after neuronal activity and contribute to removal of excess glutamate by processes called potassium buffering and glutamate clearance, respectively [12,13,14,18,19].

Mutation, downregulation or disruption of the Kir4.1 channel protein are causal in certain forms of human epilepsy [20,21,22,23]. Moreover, in humans, seizure susceptibility is increased by hyperosmolarity, hyponatremia and hyperglycemia, caused by both type 1 and type 2 diabetes [24,25]. In cell culture, we have shown that Kir4.1 mRNA and protein levels are downregulated in astrocytes cultured in high glucose medium (simulating uncontrolled diabetes) and down-regulation is associated with impaired K^+^ uptake and glutamate clearance by these astrocytes [26].

The purpose of the present study was to assess astrocytic Kir4.1 channel function and correlate this with in-vitro neuronal epileptiform-like activity in diabetic mice. We utilized the db/db type 2 diabetic mouse model which has a homozygous point mutation in the gene encoding the leptin receptor [27]. In this study, we found reduced activity of Kir4.1 channels in hippocampal astrocytes of diabetic db/db mice which was associated with deficits in potassium uptake and enhanced CA1 pyramidal cell excitability. These deficits may, in part, explain the predisposition of diabetics to suffer seizures.

## 2. Materials and Methods

### 2.1. Animals

Diabetic homozygous (db/db) and non-diabetic heterozygous (db/+) male mice were purchased from Jackson Laboratories (Bar Harbor, ME). Mice were maintained on a 12-h light/dark schedule and had access to food and water ad libitum. All experiments were approved by the Universidad Central del Caribe Institutional Animal Care and Use Committee (IACUC).

### 2.2. Blood Glucose Measurement

Mice were fasted for 5 h with access only to water. Blood was then obtained from the tail vein and fasting blood glucose levels were measured using a glucose meter and test strips.

### 2.3. SDS-PAGE and Western Blotting Analysis

The hippocampal region was dissected out of brains from control (db/+) and diabetic (db/db) mice. The samples were lysed for 60 min with RIPA buffer (Tris HCl 1.5 M pH 8.8, 1% Triton X-100, 150 mM NaCl and 0.1% SDS) with an additional mixture of peptide inhibitors (leupeptin, bestatin, pepstatin, and aprotinin) and 1.0 mM PMSF. Protein concentrations were determined using the Bradford Protein Assay (BioRad) followed by a dilution 1:3 with Urea buffer (4% SDS, 8 M Urea, 20 mM EDTA, 0.015% Bromophenol Blue, 5% β-mercaptoethanol, 62 mM Tris-HCl pH 6.8) for a final concentration of 10 µg protein/µL. Samples were resolved in 4–15% polyacrylamide gradient gels, transferred and immunoblotted using a Guinea pig polyclonal antibody against Kir4.1 (1:2000, Alomone Labs, Jerusalem, Israel), cat#AGP-035) followed by signal detection with enhanced chemiluminescence methodology (SuperSignal^®^ West Dura Extended Duration Substrate; Pierce, Rockford, IL, USA). Intensity of the signal was measured in a gel documentation system (ChemiDoc, BioRad, Hercules, CA, USA). It has been shown that Kir4.1 can appear in two forms of monomer: unglycosylated and glycosylated which correspond to 37 and 43 kDa bands, respectively [28] and we calculated the total monomer consisting of both the unglycosylated and glycosylated forms. In all cases, intensity of the chemiluminescence signal was corrected for minor differences in protein content after densitometry analysis of the India ink-stained membrane [29].

### 2.4. Electrophysiological Recording from Neurons and Astrocytes in Brain Slices

Hippocampal slices were prepared from adult male db/db and db/+ mice (90–110 days postnatal) as previously described [14]. Slices were incubated in oxygenated artificial cerebrospinal fluid (ACSF) containing: 127 mM NaCl; 2.5 mM KCl; 1.25 mM NaH_2_PO_4_; 25 mM NaHCO_3_; 2 mM CaCl_2_; 1 mM MgCl_2_; and 25 mM D-glucose for at least 1 h before recording. The solution was saturated with 95% O_2_/5% CO_2_ to achieve pH 7.4.

#### 2.4.1. Selection and Recording from Astrocytes

Astrocytes located in the CA1, exclusively in the stratum radiatum area of the hippocampus, were identified by their characteristically small rounded cell bodies and defined irregular processes and were randomly selected (attached, near and far from blood vessels) for electrophysiological recording. Astrocytes were from ~50–200 micrometer in depth of the brain slice to avoid recording from damaged cells located on both surfaces of the slices. Only passive astrocytes with a linear I/V-relationship were used for the study [14,30]. By recording from stratum radiatum, we avoided recording from satellite oligodendrocytes [31] and oligodendrocyte precursor cells [32] found in proximity to pyramidal cells. In addition, astrocytes from adult rodents have a linear response to a voltage protocol and are considered “passive glia” [30]. While a small percentage of NG2 cells also have a linear response to a voltage step protocol, the majority of these cells are “variably rectifying glia” [30]. The membrane potentials were not corrected for the theoretical junction potentials between the intracellular and extracellular solutions.

Membrane currents were measured with the single electrode whole-cell patch-clamp technique. Borosilicate glass patch pipettes (O.D. 1.5 mm, I.D. 1.0 mm; World Precision Instruments, Sarasota, FL, USA) were pulled in four-steps using a Sutter P-97 puller (Sutter Instruments, Novato, CA, USA) to a final resistance of 6–9 MΩ for astrocyte recordings. The intracellular solution contained: 138 mM KCl, 2 mM KOH, 1 mM MgCl_2_, 1 mM CaCl_2_, 10 mM EGTA, 10 mM HEPES, 1 mM spermine HCl, pH adjusted to 7.2 with KOH/HCl and had an average osmolarity of 284 ± 3 mOsm/L. After cell penetration, the access resistance was 10–18 MΩ, compensated by at least 75%.

The membrane potentials of stratum radiatum astrocytes were determined immediately after attainment of whole-cell mode, and cells were then subsequently held under voltage-clamp at this potential. Astrocytes were bathed with ACSF for 5 min to allow cell stabilization then a voltage step protocol was applied using 100 ms voltage steps to potentials between −100 mV and +100 mV from the holding potential (equal to the resting membrane potential) to measure the whole-cell currents and I/V-relationship. The patched astrocytes were then superfused with ACSF containing 100 μM barium (a selective Kir channel blocker) for 10 min after which the same voltage step protocol was applied. Barium-sensitive Kir currents were obtained by subtracting the whole-cell current in the presence of barium from the current in the absence of barium.

Using whole-cell voltage-clamp to record from astrocytes in stratum radiatum area of hippocampal slices from db/db diabetic and db/+ control mice, we measured inward K^+^ current in response to switching the external solution from one containing 2.5 mM K^+^ to one containing 10 mM K^+^ in the presence and absence of 100 μM Ba^2+^ [14,19]. Kir-dependent inward currents were obtained by subtracting the inward current in the presence of barium from the current in the absence of barium. Only one cell was recorded from each slice that was perfused with barium.

#### 2.4.2. Recording from Pyramidal Neurons

Membrane currents were measured with the single electrode whole-cell patch-clamp technique. Borosilicate glass patch pipettes (O.D. 1.5 mm, I.D. 1,0 mm; WPI, Sarasota, FL, USA) were pulled in four steps using a Sutter P-97 puller (Novato, CA, USA) to a final resistance of 2–4 MΩ for pyramidal cell recordings. The intracellular solution contained: 130 mM K-gluconate, 10 mM Na-gluconate, 4 mM NaCl, 4 mM phosphocreatine, 0.3 mM GTP-Na, 4 mM Mg-ATP, 10 mM HEPES, and the pH was adjusted to 7.2 with KOH and the average osmolarity of the intracellular solution was 279 ± 5 mOsm/L. After cell penetration, the access resistance was 9–12 MΩ, compensated by at least 75%.

The membrane potentials of CA1 pyramidal cells were determined immediately after attainment of whole-cell mode. The membrane potentials were not corrected for the theoretical junction potentials between the intracellular and extracellular solutions, because we used identical conditions for two groups of animals looking just for differences in their properties. We next recorded spontaneous action potential firing for 10 min followed by recording EPSCs for another 10 min while perfusing the slice with control ACSF solution. Action potentials were measured in gap-free current equal zero (I = 0) mode and EPSCs were measured while holding the cell at −50 mV in voltage-clamp mode [33]. Following these basal activity measurements, we induced neuronal epileptiform activity using 100 µM 4-aminopyridine (4-AP; Sigma) added to the ACSF. We measured EPSCs in the presence of ACSF with 100 μM 4-AP for 20 min and 10 min later (still in the presence of 4-AP) we determined frequency of action potential firing. Only one cell was recorded from each slice that was perfused with 4-AP.

##### Statistical Methodology

A normality diagnostic test was performed using the Shapiro-Francia estimator. Presence of outliers was verified via Grubbs test. At a bivariate level, an independent samples *t*-test was used to analyze electrophysiological data obtained from astrocytes and neuronal resting membrane potential. Variance homoscedasticity was evaluated using the Levene protocol, whereas an ordinary one-way ANOVA followed by Tukey’s multiple comparison post-hoc test was used to analyze neuronal small EPSCs. Neuronal action potential event frequency and small and large EPSC event frequency were analyzed within the same group model. This model constitutes the comparison of mice as separate units of analysis. In each case, we analyzed control (non-diabetic) groups against experimental groups in db/db mice stratified by four analytical blocks as follows: control-control; control-experimental; experimental-control and experimental-experimental., This model was analyzed with a Jonckheere–Terpstra *k* independent groups test with a modified Bonferroni post-hoc correction or a Mann–Whitney two independent groups test.

To evaluate the time effect in small EPSCs, a modified General Linear Model Repeated Measures ANOVA was used. A Mauchly’s test of sphericity was performed to assess if our model has the assumption of compound symmetry. If non-significant (*p* ≤ 0.05) we report the univariate results with a Greenhouse–Geisser epsilon correction; if significant (*p* < 0.05), we report the multivariate results using the Pillai’s trace estimator. Either of the last explained results was used to evaluate the time effect in our models.

The estimated marginal means with their correspondent standard errors are reported for each factor in the experiments. The significant level (α) was set to ≤0.05, excluding the normality and homocedasticity tests (*p* > 0.05). The IBM Statistical Package for Social Sciences v.23.0 for Windows was used (IBM-SPSS, Chicago, IL, USA).

## 3. Results

We have previously shown that normal astrocytes cultured in hyperglycemic conditions switched their behavior and have impaired function of Kir4.1 channels and decreased capability to buffer extracellular glutamate [26]. We now extended these findings to examine if Kir4.1 channel protein expression is decreased in hippocampus of diabetic db/db male mice (*n* = 3) as compared with non-diabetic db/+ controls (*n* = 3). The average fasting blood glucose levels were 138 ± 23 mg/dL (mean ± SEM; *n* = 3) for control mice and 547 ± 30 mg/dL for diabetic mice. In the case of one db/db mouse the blood glucose level was higher than the sensitivity of the meter and we used the upper detectable level of 600 mg/dL for this mouse. Using Western blot analysis (Figure 1), we obtained a 43 kDa band corresponding to glycosylated Kir4.1 monomer and a 37 kDa band corresponding to the unglycosylated Kir4.1 monomer as previously described [28]. Using both bands for calculating Kir4.1 monomer levels in control and diabetic mice, we found that protein levels were 31% lower in the hippocampal region of db/db diabetic mice as compared with db/+ (Figure 1). Kir4.1 is found primarily in glial cells in brain [11,33], therefore, these data reflect a reduction in Kir4.1 channels in glia, not neurons.

Furthermore, we examined if Kir4.1 channel electrophysiological properties are impaired in diabetic db/db male mice as compared with non-diabetic db/+ controls. The fasting blood glucose levels for db/+ control mice ranged between 123 mg/dL and 143 mg/dL, whereas the levels for db/db diabetic mice were in the range of 425 mg/dL to 513 mg/dL.

Since the highly hyperpolarized membrane potential of astrocytes is largely due to the expression of Kir4.1 channels, we measured the membrane potential of astrocytes in CA1 hippocampal brain slices obtained from diabetic db/db and non-diabetic db/+ mice. Astrocytes from db/db mice were depolarized (−78.0 ± 1.3 mV; *n* = 53 from 7 mice) when compared to their db/+ counterparts (−85.7 ± 0.5 mV; *n* = 51 from 9 mice) (Figure 2A). Some of the membrane potentials of astrocytes from db/db mice were highly hyperpolarized as for those of db/+ astrocytes, but a population of the cells in db/db mice were more depolarized. The difference was reflected in the median membrane potentials for astrocytes from diabetic and control mice which were -80mV and -86mV, respectively and can be clearly seen in the scatterplot (Figure 2A).

To determine the contribution of Kir channels to the total whole cell currents of CA1 hippocampal astrocytes, brain slices were first perfused with a standard ACSF solution and astrocytes were clamped and held at their native resting membrane potential (Vm), so that Vm = Vh. Under these conditions, application of 100 ms voltage steps to potentials between −100 mV and +100 mV from the holding potential (equal to the resting membrane potential) evoked both inward and outward currents that were substantially larger in db/+ astrocytes than those elicited in db/db astrocytes (Figure 3A1). Upon application of the Kir channel blocker, Ba^2+^ (100 μM), currents in astrocytes from db/+ mice were inhibited by about 50% (Figure 3A2 open squares), but there was significantly less effect of Ba^2+^ on the current-voltage curve of db/db astrocytes (Figure 3A2, filled blue squares,) as summarized in Figure 3C. To obtain the Ba^2+^ sensitive currents which reflect the current contributed by functional Kir channel expression (Figure 3A3), we subtracted the current in the presence of Ba^2+^ (Figure 3A2) from the total whole currents (Figure 3A1). The average inward whole cell currents measured at −150 mV in ACSF are shown in Figure 3B. Ba^2+^ sensitive inward currents (measured at −150 mV) from diabetic mice astrocytes (−0.4 nA; *n* = 21 cells from 7 mice) were substantially smaller than those obtained from non-diabetic animals (−1.8 nA; *n* = 11 cells from 9 mice) and are summarized as percent of current blocked by barium in Figure 3C. Taken together, the depolarized membrane potential, the smaller current elicited from db/db astrocytes and the decreased response to Ba^2+^ application indicate that functional (membrane) Kir channel activity is substantially reduced in astrocytes of db/db mice as compared with db/+ astrocytes.

One of the major functions of astrocytes is the removal of excess potassium from active synaptic areas [11,18]. Kir4.1 channels in astrocytes contribute to potassium uptake, therefore, we tested the potassium uptake capabilities of astrocytes via Kir channels using a physiologically relevant protocol [12,14,19,26]. We clamped each astrocyte to their native resting membrane potential, thus zero current, while perfusing the slice with control ACSF containing 2.5 mM K^+^. We then measured inward K^+^ current in response to switching the external solution from one containing 2.5 mM K^+^ to one containing 10 mM K^+^ (Figure 4A). This was done in the presence and absence of 100 μM Ba^2+^. Average inward currents generated by switching extracellular K^+^ from 2.5 to 10 mM were significantly smaller in astrocytes from diabetic mice (−403 ± −102 pA; *n* = 12 cells from 6 mice) compared to astrocytes from non-diabetic mice (−668 ± −101 pA; *n* = 15 cells from 6 mice). In addition, we found that there was significantly less Ba^2+^ sensitive Kir current recorded in astrocytes from diabetic mice (26 ± 3%; *n* = 15 cells from 6 mice) compared with non-diabetic mice (39 ± 5%; *n* = 12 cells from 6 mice; Figure 4B).

Electrophysiological properties of hippocampal CA1 pyramidal neurons were measured in brain slices from diabetic and non-diabetic mice. We first measured the resting membrane potential of pyramidal cells in the CA1 area of the hippocampus. As shown in Figure 2B, the resting membrane potentials of CA1 pyramidal cells of non-diabetics (−67 ± 1.5 mV; *n* = 22 cells from 8 mice) and diabetics (−67 ± 1.4 mV; *n* = 22 cells from 5 mice) were not different.

Spontaneous action potential activity was determined for both diabetic and non-diabetic mice (Figure 5). Interestingly, 77% of CA1 pyramidal cells recorded from diabetic mice and only 39% of the CA1 pyramidal cells from non-diabetic mice displayed spontaneous action potential firing in brain slices (Figure 5A). Similar results were seen after application of the pro-convulsant 4-aminopyridine (4-AP; 100 µM). Thirty minutes after application of 4-AP, 69% of the CA1 cells from diabetic animals and 42% of the recorded cells from non-diabetic mice were spontaneously active (Figure 5A).

We next measured the action potential frequency (averaged over 10 min) before and after application of 4-AP. Action potential frequency was ≈9 times higher in diabetic (0.26 ± 0.12 Hz; *n* = 13 cells from 5 mice) compared with non-diabetic mice (0.03 ± 0.01 Hz; *n* = 11 cells from 8 mice) prior to application of 4-AP (Figure 5B). Furthermore, 30 min after perfusion with 100 µM 4-AP, the average action potential frequency increased for CA1 pyramidal cells recorded from slices obtained from both diabetic (0.69 ± 0.36 Hz; *n* = 11 cells from 5 mice) and non-diabetic mice (0.15 ± 0.07 Hz; *n* = 11 cells from 8 mice). In addition, the firing frequency was significantly greater (≈5 times) for CA1 neurons from diabetic mice compared with control mice (Figure 5B).

Excitatory post-synaptic currents (EPSC) were recorded from CA1 pyramidal cells in the presence and absence of 4-AP (100 µM), while holding the cells at −50 mV. EPSCs were classified as small or large based on the amplitude of the recorded inward current. Small EPSCs were 20 pA or smaller whereas currents greater than 20 pA were categorized as large EPSCs. The small EPSCs were counted manually for one minute every 5 min of recording, whereas the large EPSCs were calculated using the event detection and threshold search feature of clampfit (Axon Instruments) set as 21 pA. Figure 6A (control) and 5B (diabetic) show the percent of cells displaying any small or large EPSCs before or after application of 4-AP. Less than 20% of the cells from non-diabetic mice displayed small EPSCs in ACSF whereas over 40% of the cells recorded from diabetic mice showed small EPSCs while superfused with ACSF. As expected, the number of cells displaying EPSCs increased after application of 4-AP in slices from both non-diabetic and diabetic mice (Figure 6A,B).

We then determined frequency of small EPSCs in one minute intervals every 5 min (Figure 7A). Interestingly, we found that the frequency of EPSCs was significantly higher in CA1 neurons from diabetic mice throughout the entire recording period (i.e., both before and after 4-AP) and this difference was enhanced after the application of 4-AP (Figure 7A). Indeed, after application of 100 µM 4-AP, the small EPSC frequency was significantly increased only in CA1 pyramidal cells in slices from diabetic mice, but not for non-diabetic mice. This provides further indication that neurons from diabetic mice are hyperexcitable compared to control.

Finally, we measured large EPSC frequency in one minute intervals every 5 min for 10 min before application of 4-AP and for 20 min after application of 4-AP (Figure 7B). There was no significant difference in the large EPSC frequency observed in CA1 neurons between diabetic (*n* = 12 cells from 5 mice) and non-diabetic mice (*n* = 11 cells from 8 mice) prior or after application of 4-AP. Large EPSCs have been reported to be the summation of synchronized inputs onto the cell and they become particularly apparent after 4-AP. After 4-AP, the frequency of large EPSCs was significantly increased in both diabetic and non-diabetic mice.

## 4. Discussion

While the association between seizures and diabetes is widely accepted in the medical community, the pathophysiology has yet to be elucidated [25,34,35]. It has been shown that long-term treatment with streptozotocin to induce hyperglycemia/diabetes increases epileptiform like activity in pyramidal neurons in the CA3 area of the hippocampus [36]. Moreover, streptozotocin treated rats display increased seizure severity and death related to status epilepticus induced by pilocarpine [37]. In addition, these rats show markedly increased neuronal loss and reduced long-term potentiation in the CA3 area of the hippocampus [37]. Consistent with these findings, systemic administration of glucose in rats (hyperglycemic condition) decreased seizure threshold in the flurothyl seizure test [38].

To date, there are no reports about spontaneous seizure activity in db/db mice. Therefore, in the present study, we used a db/db mouse model of type 2 diabetes [27] and evaluated neuronal excitability in both basal conditions and after application of 4-AP. We compared the electrophysiological properties of CA1 hippocampal neurons in brain slices obtained from non-diabetic db/+ mice and diabetic db/db mice. Although there was no difference in the mean resting membrane potential of CA1 pyramidal cells from non-diabetic and diabetic mice, neurons recorded from diabetic mice were hyperexcitable. This was evident in both spontaneous action potential firing as well as the frequency of small EPSCs with both being higher in diabetic mice.

There are a number of intrinsic mechanisms reported that have been shown to contribute to increased neuronal excitability such as: (i) increased number and distribution of ion channels [39], (ii) activation of second-messenger systems that affect channel function [40] and (iii) modulation of gene expression of ion channels [41]. However, there are also a number of extrinsic mechanisms that could influence neuronal excitability such as: (i) changes in extracellular ion concentration [42], (ii) remodeling of synaptic contacts [43] and (iii) modulation of transmitter metabolism or uptake by glial cells [44]. The present study focuses on the potential influence of extrinsic mechanisms to alter neuronal excitability.

The db/db mice are hyperglycemic. Among the effects of hyperglycemia seen on neurons in streptozotocin-induced diabetic rats are increased α-amino-3-hydroxy-5-methyl-4-isoxazolepropionic acid (AMPA) and NMDA receptor channel density [45,46]. However, type 2 diabetic db/db mice also have increased plasma insulin levels and it has been reported that insulin, by itself, could positively potentiate current through NMDA receptors and increase neuronal electrical coupling [46].

Furthermore, among the effects of hyperglycemia seen in glial cells, It has been shown that GFAP immunoreactivity increases in the hippocampus of streptozotocin treated mice [47]. In addition, there is an increase in the numbers of GFAP^+^ Müller cells in the retina of db/db mice as compared with non-diabetic controls [48]. Additionally, it is known that hyperglycemia reduces Kir4.1 potassium channel expression and activity in cultured cortical astrocytes [26] and in retinal Müller glial cells [49]. Downregulation of Kir4.1 potassium channels in astrocytes may provide a compelling mechanism to account for the propensity of diabetics to have seizures. Key features of astrocytes that make them well suited to maintain ionic and neurotransmitter homeostasis are their large K^+^ permeability and their highly hyperpolarized native membrane potential.

One of the main contributors of a hyperpolarized membrane potential is the functional expression of Kir4.1 channels in astrocytic membranes. The strict dependence of the Kir4.1 channels on voltage and extracellular K^+^ variations allows K^+^ influx at hyperpolarized resting membrane potential and also efflux when [K^+^]_o_ is low [11,50]. Natively, they can be found as homomeric channels or as heteromeric channels together with Kir5.1 subunits. However, in the hippocampus, the Kir4.1 homomer predominates [17,28].

Although Kir4.1 channels are the major channels in astrocytes which contribute to resting membrane potential and K^+^ uptake [12,13,17], there are other K^+^ channels that may contribute to these processes in astrocytes and other glial cells. Seifert et al., (2009) showed that hippocampal astrocytes express not only Kir4.1 channels, but also express other Kir channel subunits from the Kir2.0 family [17] although only the Kir4.1 transcript was invariably found in all the astrocytes tested. Thomzig et al., (2001) reported that hippocampal astrocytes also express the Kir6.1 pore-forming subunit [51], but these Kir6.1 containing channels only function when cells are metabolically challenged and ATP concentrations are reduced. In addition to Kir channels, astrocytes have been reported to express tandem pore domain K^+^ channels including TREK-1, TREK-2, and TWIK-1 [17,52,53] as well as TASK-1 and TASK-3 [52]. Some of these channels may contribute to the barium-insensitive component of the resting membrane potential and K^+^ uptake by astrocytes.

In addition, it has been shown that downregulation of Kir4.1 channels affects overall ion gradients and also impairs glutamate uptake by astrocytes [12,13]. There are numerous theoretical analyses supporting the spatial relationship between passive and active modulation of [K^+^]_o_ by astrocytes and the initiation or maintenance of epileptiform activity [54,55,56]. Furthermore, cell depolarization and impaired ion gradients inhibit glutamate transporter function and cause transporter reversal, thus glutamate release from the cell, which could further affect neuronal excitability [57,58].

Therefore, dysfunction of astrocytic Kir4.1 channels may alter extracellular brain homeostasis and affect neuronal activity. Indeed, mutations and variations of the KCNJ10 gene encoding for Kir4.1 occur in seizure susceptible human and animal models [14,59,60] including the SeSAME/EAST syndrome [20,21,23]. In seizure susceptible DBA mice, astrocytic Kir currents are reduced and this has been correlated with decreased K^+^ uptake and glutamate clearance by these cells [14]. Other studies have shown Kir4.1 downregulation as well as impaired glutamate and potassium uptake when glial cells are exposed to hyperglycemic/diabetic conditions in both glial cells in culture [26] and in streptozotocin-induced diabetic animals [49]. In the present study, we found that astrocytes in hippocampal slices from diabetic mice had a depolarized membrane potential and reduced functional Kir channel activity. It is well established that astrocytes regulate extracellular K^+^ concentration by taking up excess K^+^ from active synaptic areas and by redistributing it to sites of lower K^+^ concentration through the astrocytic syncytium [16,18].

When an action potential propagates, extracellular K^+^ increases [61]. If this excess potassium is not removed, it could cause hyperexcitability, seizures and neuronal death. Therefore, we evaluated the ability of astrocytes to take up excess K^+^ by stepping external K^+^ from 2.5 mM to 10 mM and measuring the inward current generated. As predicted, we found that astrocytes from diabetic animals displayed impaired K^+^ uptake capabilities compared with non-diabetic mice. Taken together, reduced astrocytic membrane potential, reduced Kir channel activity, and impaired K^+^ uptake in diabetic mice could impact the ability of astrocytes to maintain extracellular ion homeostasis and consequently affect neuronal excitability.

## 5. Conclusions

In conclusion, we examined the association between CA1 pyramidal cell membrane excitability together with astrocytic Kir4.1 potassium channel function in the hippocampus using a type 2 diabetic mouse model. We found that CA1 pyramidal neurons from diabetic mice were hyperexcitable and this correlated with reduced functional Kir4.1 channel activity in astrocytes. Overall, our data provide insight into a potential Kir4.1 potassium channel-dependent mechanism that may help explain the high incidence of seizures seen in patients with uncontrolled hyperglycemia.

## Figures and Tables

**Figure 1 brainsci-10-00072-f001:**
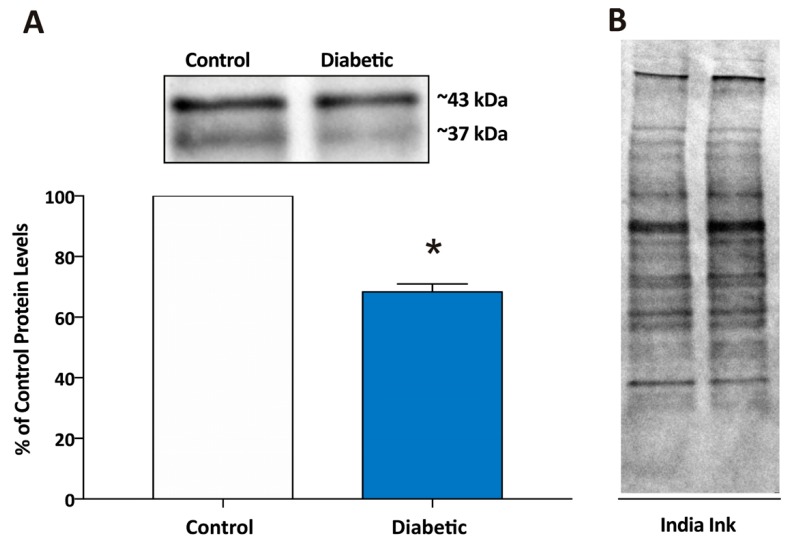
Kir4.1 potassium channel protein levels are lower in the hippocampal region of brains from diabetic mice as compared with non-diabetic control mice. (**A**) Kir4.1 potassium channel protein levels (total monomer of both glycosylated and unglycosylated forms of Kir4.1) measured by Western Blot in hippocampus from diabetic mice were significantly downregulated (68.3 ± 2.6%, *n* = 3) as compared to control. Data are expressed as % of control with control being hippocampus from non-diabetic mice and * indicating a significant difference from control group (*p* < 0.05; Student’s *t*-Test for independent samples). Above the graph in (**A**) are representative bands from the Western blot showing Kir4.1 protein levels from a control and a diabetic mouse. The band detected at ~37 kDa corresponds to the unglycosylated form, whereas the band detected at ~43 corresponds to the glycosylated form of the Kir4.1 monomer. (**B**) The India ink-stained membrane showing the total protein of the samples shown in (**A**) that was used for loading control calculations.

**Figure 2 brainsci-10-00072-f002:**
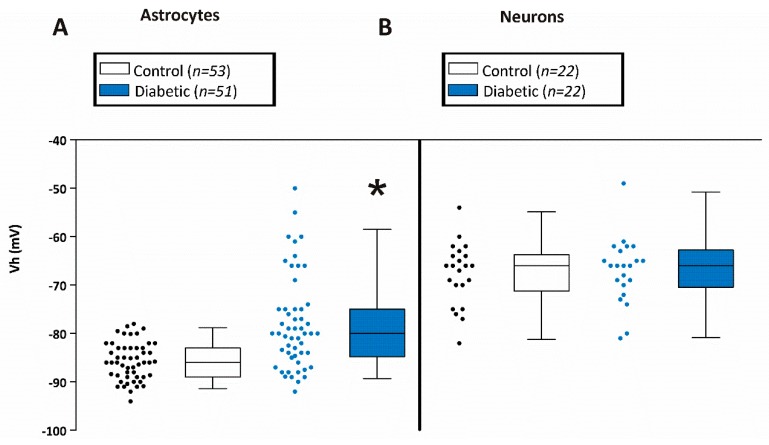
Membrane potentials of astrocytes and neurons in CA1 hippocampal slices from control and diabetic mice. (**A**) Average membrane potential of astrocytes from hippocampal CA1 stratum radiatum area of brain slices from control (−85.7 ± 0.6 mV; mean±SEM; *n* = 51 from 9 mice) and diabetic (−78.0 ± 1.3 mV; *n* = 51 from 7 mice) mice. Asterisk * indicates significant difference from the control group (*p* < 0.05; Student’s *t*-Test for independent samples). (**B**) Average membrane potential of pyramidal neurons from hippocampal CA1 stratum radiatum area from control (−67 ± 1.4 mV; *n* = 22 from 8 mice) and diabetic (−67 ± 1.5 mV; *n* = 22 from 5 mice) mice.

**Figure 3 brainsci-10-00072-f003:**
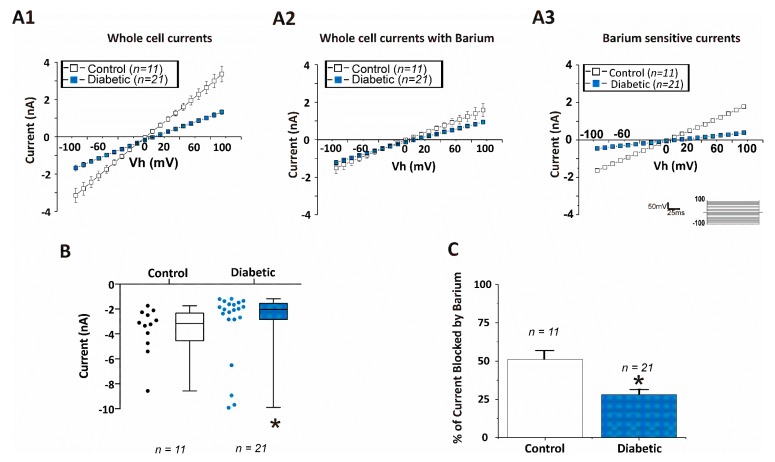
Kir currents are reduced in CA1 hippocampal astrocytes from diabetic mice. Astrocytes were clamped at holding potential (Vh) which was equal to the resting membrane potential (Vm) in 2.5 mM [K^+^]_o_ containing artificial cerebrospinal fluid (ACSF), Vh = Vm. I/V-curves are shown in response to a voltage step protocol (100 ms voltage steps to potentials between −100 mV and +100 mV from the holding potential (Vh)). (**A1**) Whole cell currents recorded in response to a voltage step protocol from astrocytes in hippocampal brain slices obtained from control (open squares) or diabetic (filled blue squares) mice. (**A2**) Whole-cell currents recorded in response to a voltage step protocol in the presence of 100 μM Ba^2+^ (a blocker of Kir channels) from astrocytes in hippocampal brain slices obtained from control or diabetic mice. (**A3**) Ba^2+^ sensitive Kir currents from astrocytes from control or diabetic mice. The graph shows the subtraction of currents obtained in the presence of Ba^2+^ (**A2**) from total whole-cell currents shown in (**A1**). Ba^2+^-sensitive currents reflect the contribution of Kir channels to the whole cell currents. (**B**) Inward current measured at −150 mV in astrocytes from control and diabetic mice. (**C**) Percentage of current blocked by 100 μM Ba^2+^ in astrocytes from control and diabetic mice. Data were obtained from 9 db/+ control and 7 db/db diabetic mice. Data in **(B**,**C**) are displayed as mean±SEM. Asterisk (*) indicates significant difference from the non-diabetic control group (*p* < 0.05; Student’s *t*-Test for independent samples).

**Figure 4 brainsci-10-00072-f004:**
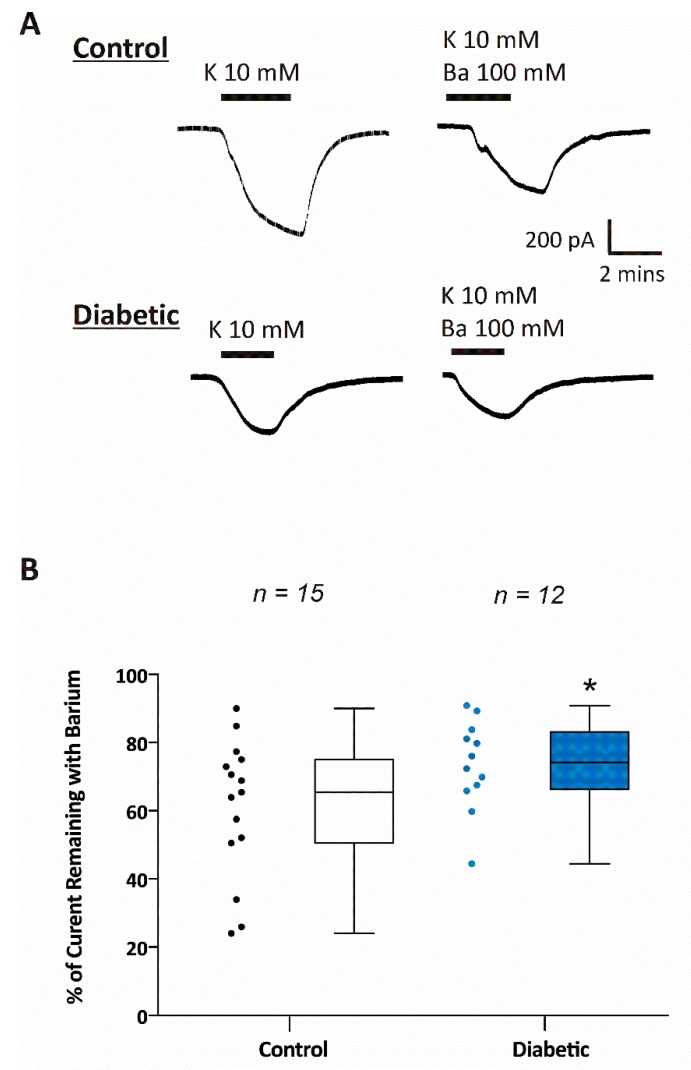
Potassium uptake is impaired in CA1 hippocampal astrocytes from diabetic mice. (**A**) Representative whole cell currents recorded from astrocytes of control and diabetic mice in hippocampal brain slices. Inward currents were obtained by changing extracellular K^+^ from 2.5 mM to 10 mM in the presence or absence of 100 μM Ba^2+^. The cells were held at the steady state potential (Vh = Vm). The scales bars are equal for all current traces. (**B**) Summary of the relative barium insensitive Kir currents measured in hippocampal astrocytes from control (*n* = 15) and diabetic (*n* = 12) mouse brain slices. In **(B),** the data are expressed as percent (%) of control current that is barium sensitive where 100% is the maximal current measured by switching extracellular [K^+^] from 2.5 to 10 mM. Data were obtained from 6 db/+ control and 6 db/db diabetic mice. Asterisk (*) indicates significant difference from the non-diabetic control group (*p* < 0.05; Student’s *t*-Test for independent samples).

**Figure 5 brainsci-10-00072-f005:**
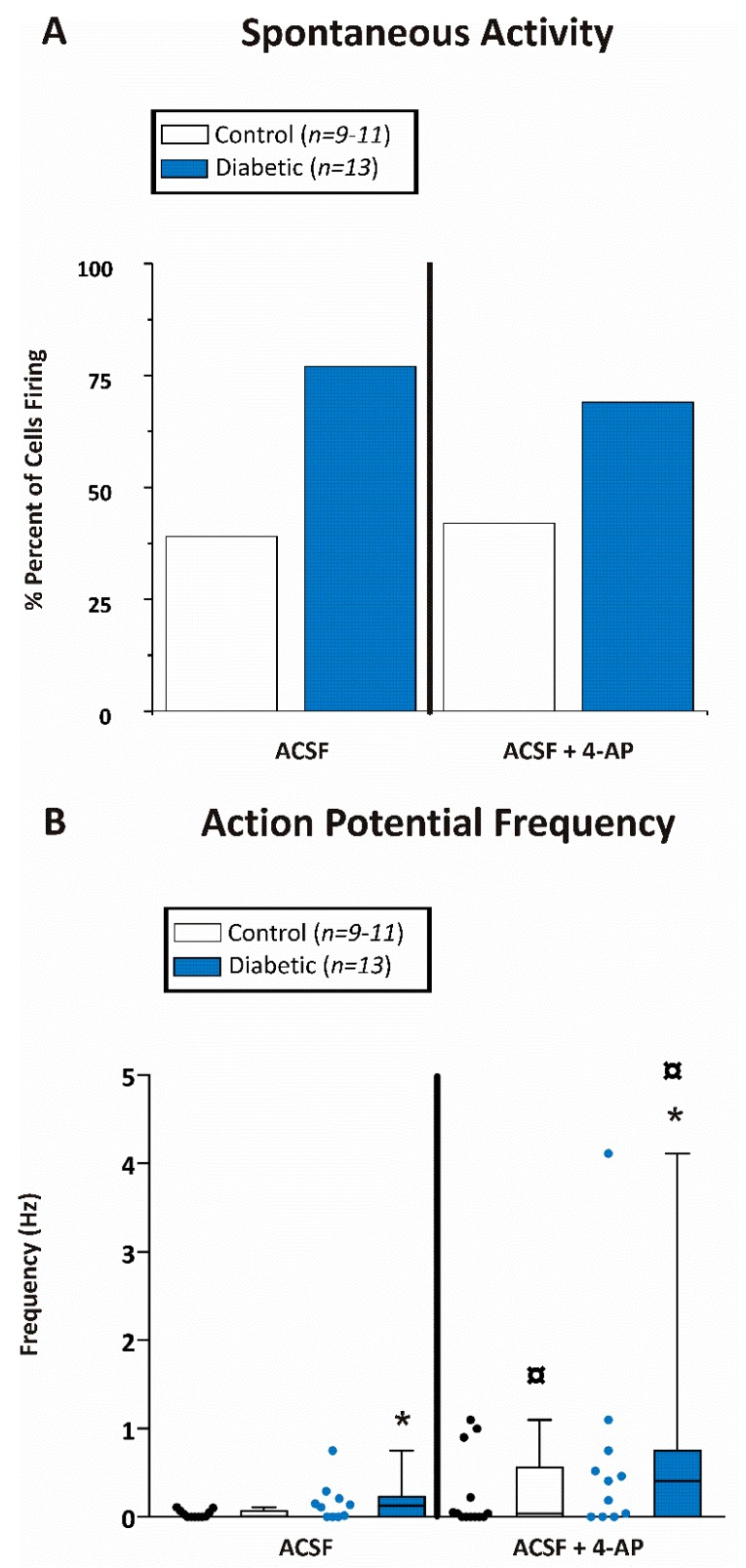
CA1 pyramidal neurons from diabetic mice were hyperexcitable. (**A**) Percent of CA1 pyramidal neurons firing any action potentials within the 10 min recording period while being perfused with either ACSF or ACSF containing 100 µM 4-aminopyridine (4-AP). (**B**) Action potential firing frequency (Hz) of CA1 pyramidal neurons during the 10 min recording period while being perfused with either ACSF or ACSF containing 100 µM 4-AP. Data were obtained from 9-11 cells of 8 db/+ control and 13 cells from 5 db/db diabetic mice. Asterisk (*) indicates significant difference from the corresponding non-diabetic control whereas the boxed circle (
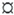
) indicates significant difference after application of 4-AP from the same non-diabetic or diabetic ACSF group (*p* < 0.05; Jonckheere–Terpstra *k* independent groups test).

**Figure 6 brainsci-10-00072-f006:**
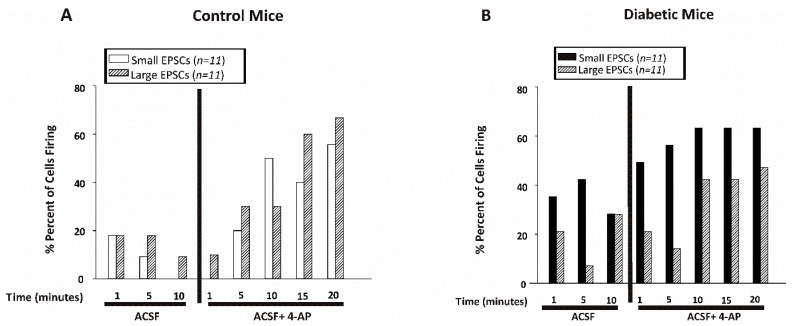
CA1 pyramidal neurons from diabetic mice displayed more small excitatory post-synaptic currents. Percent of cells displaying small EPSCs (≤20 pA) or large EPSCs (>20 pA) recorded for one minute every 5 min of recording in CA1 pyramidal neurons from control (**A**) or diabetic (**B**) mice. Data were obtained from 11 cells from 8 db/+ control and 12 cells from 5 db/db diabetic mice.

**Figure 7 brainsci-10-00072-f007:**
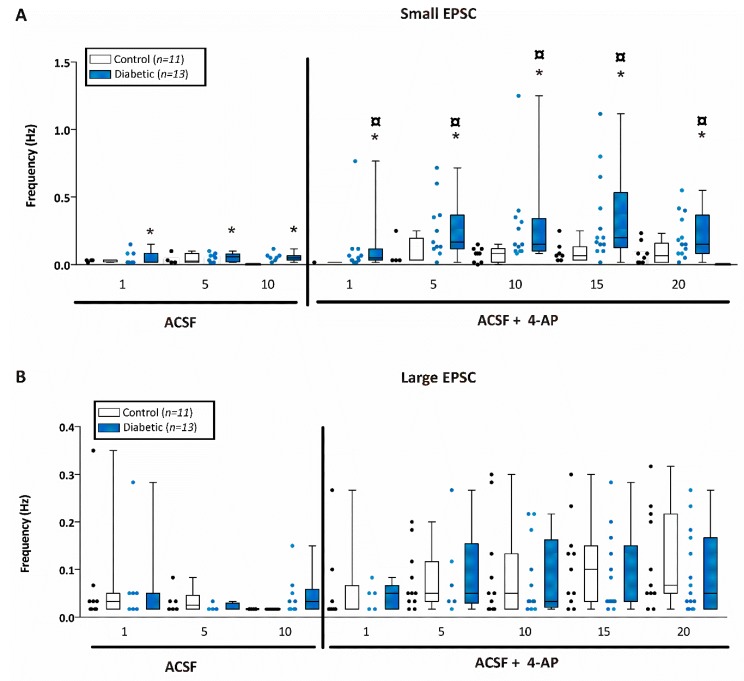
CA1 pyramidal neurons from diabetic mice displayed more excitatory post-synaptic currents. Excitatory post-synaptic currents (EPSC) event frequency of small EPSCs (≤20 pA) or large EPSCs (>20 pA) recorded for one minute every 5 min of recording in CA1 pyramidal neurons from control or diabetic mice. (**A**) Small EPSC frequency (Hz) recorded in CA1 pyramidal neurons from control (open bars) or diabetic (filled blue bars) mice before and after application of 4-AP. Asterisk (*) indicates significant difference from the corresponding non-diabetic control whereas boxed circle (
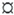
) indicates significant difference after application of 4-AP from the same non-diabetic or diabetic ACSF group (*p* < 0.05; Jonckheere–Terpstra k independent groups test). (**B**) Large EPSC frequency (Hz) recorded in CA1 pyramidal neurons from control (open bars) or diabetic (filled blue bars) mice. Data were obtained from 11 cells from 8 db/+ control and 12 cells from 5 db/db diabetic mice. The large EPSCs recorded from CA1 pyramidal neurons were significantly greater after 4-AP in both control and diabetic mice although there was no difference in the frequency between control and diabetic mice at any time point examined either before or after 4-AP.

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
