# Peer review of "Downregulation of Astrocytic Kir4.1 Potassium Channels Is Associated with Hippocampal Neuronal Hyperexcitability in Type 2 Diabetic Mice"

_brainsci, 2020, doi:10.3390/brainsci10020072_

Round 1
Comments and Suggestions for Authors
“It would be useful to show either by WB, qPCR or immunohistochemistry that Kir4.1
expression is downregulated in the hippocampus of db/db mice relative to WT littermates.
We now include additional data by WB demonstrating that Kir4.1 protein expression is reduced in
hippocampus of diabetic (db/db) mice as compared with control (db/+) mice. These new data are
included in the manuscript as the new figure 1.
“In figure 3- caution should be used when comparing experiment performed to K+ The
condition used-bath applying high K+ - do not allow for K+ buffering, but rather uptake.”
We have carefully checked throughout the manuscript and fixed/reworded when needed to
properly address this.
“As it relates to figure 3- are currents remaining in the presence of Ba2+ mediated by other
astrocyte K+ channels or could they potentially be inhibited by simply increasing the Ba2+
concentration.” “This may be important because if the uptake current is not Ba2+ sensitive
Kir, the results indicate that other astrocyte K+ channels may also contribute to RMP and
uptake capabilities. At a minimum, please discuss.”
Although Kir4.1 channels are the predominate channels in astrocytes which contribute to RMP
and uptake capabilities (Djukic et al., 2007; Kucheryayvkh et al., 2007; Seifert et al., 2009), there
are other K+ channels that may contribute to these processes in astrocytes and other glial cells.
Seifert et al., (2009) showed that hippocampal astrocytes express not only Kir4.1 channels, but
also express other Kir channel subunits including the Kir2.0 family subunits (Seifert et al., 2009)
as well as the Kir5.1 subunit (which forms a heteromeric channel with Kir4.1). Of the Kir subunit
transcripts found in hippocampal astrocytes, only the Kir4.1 subunit was invariably found in all the
astrocytes tested. Thomzig et al., (2001) reported that hippocampal astrocytes also express the
Kir6.1 pore forming subunit, but these Kir6.1 containing channels only function when cells are
metabolically challenged and ATP concentrations are reduced. In addition to Kir channels,
astrocytes have been reported to express tandem pore domain K+ channels including TREK-1,
TREK-2, and TWIK-1 (Seifert et al., 2009; Kucheryavykh et al., 2009; Zhou et al., 2009) as well
as TASK-1 and TASK-3 (Kucheryavykh et al., 2009). Some of these channels may contribute to
the K+ uptake capabilities and resting membrane potentials of astrocytes.
For example, in retinal Müller glial cells in which Kir, KA, KD and BK channels were blocked,
Skatchkov et al., (2006), showed that there was still a K+ conductance due to tandem pore domain
K+ channels which could contribute to the RMP especially when Kir4.1 was blocked. On the other
hand, Du et al., (2016) demonstrated that genetic knockout of TREK-1 and TWIK-1 K+ channels
did not affect the basic electrophysiological properties of hippocampal astrocytes. Thereby, TWIK-
1/TREK-1 have a very limited influence of electrophysiology of astrocytes.
We have now added a few sentences in the discussion about the other K+ channels in astrocytes
that may contribute to RMP and K+ uptake.
“Spontaneous activity in neurons should be analyzed using spontaneous activity software
such as synaptosoft (can be used in free demo mode) or one of many others.”
We downloaded the Synaptosoft demo software and tried to analyze our files using this software.
It is theoretically possible, but we seem to have compatibility issues and would have to convert
all of the files recorded to a compatible extension. Although the Synaptosoft software may provide
additional information such as the actual amplitudes of the EPSCs, there should be no difference
in the major results obtained using the event detection and threshold search feature of Clampfit
instead of the suggested software (i.e., increased neuronal hyperexcitability in diabetic mice).
If the reviewer feels strongly that additional analysis should be done with a spontaneous activity
software, we would need additional time to analyze the numerous data files and modify the figures
and manuscript.
Minor:
“Please indicate if db/db mice have been reported to have seizures.”
To date, there are no published reports about spontaneous seizure activity in db/db mice.
“Please indicate if it is known if there is reactive gliosis in db/db mice or other models of
diabetes”
It has been shown that GFAP immunoreactivity increases in the hippocampus of Streptozotocin
treated mice; a model of hyperglycemia (Wanroov et al., 2018). In addition, there is an increase
in the numbers of GFAP+ Müller cells in the retina of db/db mice as compared with non-diabetic
controls (Bogdanov et al., 2014).
This information has been added to the discussion section of the manuscript.
“Line 385 – 12 mM is the ceiling of K+ recorded, and it is unlikely this concentration is
reached in a non-pathological context. Please consider rewording or removing this
sentence.”
As suggested, we have modified this sentence.
Round 2
The authors have addressed my concerns in the revised manuscript.
---------------------------------------
Reviewer 2 Report
Reviewer Two Comments
Comments and Suggestions for Authors
Dear Authors,
“Your paper is well thought out, well organized and very well crafted.
I have questions in regards to extrinsic mechanisms of remodeling of the synaptic
contacts. In the db/db models there is activation of microglia and I have noted that the
microglia can invade these synapses and either promote a necessary positive pruning or
it can actually cause a negative effect that being the detachment and retraction of the AC
from the tripartite synapse. Do you have any thoughts regarding this remodeling in the
db/db model. Also, do you have any thoughts regarding the influence of known insulin
and leptin resistance in the db/db models in regard to your functional observations and
findings?
While these questions are important to this reviewer, it is not necessary to discuss these
possibilities in the submitted manuscript.”
We have previously shown that astrocytes cultured in hyperglycemic conditions have
downregulation of the Kir4.1 channel protein and mRNA as compared with astrocytes cultured in
lower glucose. Since these changes occur in a culture system that does not depend on the insulin
and leptin resistance of the db/db models, we attribute these changes to exposure to
hyperglycemia.
That being said, the db/db mouse model is much more complex and certainly more than just
Kir4.1 is altered in the brains of these mice. In addition to the hyperglycemia, the mice also have
insulin and leptin resistance and are morbidly obese which may have additional influences on the
synaptic circuitry of the hippocampus. As noted above, activation of microglia and subsequent
pruning of synapses could indeed alter the tripartite synapse. This is something we have not
tested in the current manuscript. Furthermore, astrocytes express insulin receptors (García-
Cáceres, et al., 2016) and changes in astrocytic insulin sensitivity and brain insulin signaling may
be modified in these mice. In the future, we plan to assess the contribution of downregulation of
Kir4.1 in astrocytes to our functional observations and findings about neuronal hyperexcitability
by reinstating Kir4.1 in astrocytes using an AAV-Kir4.1 that selectively targets expression in glial
cells (Tong et al., 2014).
Thank you for your insightful discussion about other potential mechanisms.
Round 2
Thank you for making the necessary changes as per recommendations.